# Hidden Blemish in European Law: Judgements on Unconventional Monetary Programmes

**Bodo Herzog** [†,‡]

ESB Business School, Reutlingen University, 72762 Reutlingen, Germany; Bodo.Herzog@Reutlingen-University.de
† RRI—Reutlingen Research Institute.
‡ IFE—Institute of Finance and Economics.

**Abstract:** This article studies the hidden blemishes of two benchmark rulings of the European Court of Justice (ECJ). In 2015 and 2018, the ECJ approved two unconventional monetary instruments, among others 'Outright Monetary Transactions' and the 'Public Sector Purchase Program'. Yet, there is a vigorous debate about both monetary operations in law and economics. In this interdisciplinary article, we address law and economic arguments in order to elucidate insights to the legal community. In particular, we elaborate on the legal implications of a variety of concerning issues such as public policy interference, effect on wealth redistribution, erosion of democratic legitimacy and lack of effectiveness of monetary policy. These topics remain disregarded in the ECJ rulings. Consequently, the verdicts do not identify the economic boundaries of the European Central Bank's mandate appropriately.

**Keywords:** European Court of Justice; Bundesverfassungsgericht; European Central Bank; European union law; German constitutional law; ultra-vires; monetary policy; unconventional programmes; PSPP; OMT



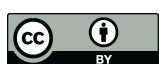

## 1. Introduction

On 16 June 2015, the European Court of Justice (ECJ) concluded that 'Outright Monetary Transactions' (OMTs) of the European Central Bank (ECB) are consistent with European law. Similarly, on 11 December 2018, the ECJ approved the 'Public Sector Purchase Program' (PSPP). In both cases the court simply followed the ECB's rationale that unconventional monetary programs are necessary in order to guarantee the singleness of the monetary union (ECJ 2015, Rn 66, 67). This interpretation broadens the ECB's monetary mandate further in regard to the 'whatever-it-takes' philosophy. However, the assigned broad discretion of the mandate creates additional fuzziness to the treaty provisions and more importantly weakens the European fiscal architecture in the end.

The cornerstone of Europe's fiscal architecture is the Stability and Growth Pact[1] supplemented with the no-bailout rule in Article 125 on the 'Treaty of the Function of the European Union' (TFEU). In a supranational monetary union, fiscal governance is required in order to ensure sustainable public finances and thus avoid free-riding and moral hazard. Nonetheless, even the present fiscal architecture has flaws and lacks cohesive forces (Herzog and Hengstermann 2013).

This article elaborates mainly upon the legal transgression of unconventional monetary policy due to vague treaty provisions. We ask to what extent unconventional operations transgress the fiscal and economic sovereignty of Member States. The ECJ states that as long as the unconventional instrument pursues the primary objective of maintaining price-stability in Article 127(1) TFEU, all programmes fall within the ambit of

---

[1] Stability and Growth Pact Council Regulation (EC) 1466/97 and 1467/97 and the updates 1055/05 and 1056/05 and recently Regulations (EU) No. 1173/2011, No. 1174/2011, No. 1176/2011, and No. 472/2013 and No. 473/2013.

the ECB (ECJ 2018, Rn 45, 51). However, our analysis reveals several confines because the verdict remains incomplete and imprecise.

Generally, there are two opposing views about the functionality of unconventional instruments. On the one hand, unconventional policies give Member States more time in order to impose policy reforms. This facilitates economic growth and lifts inflation. On the other hand, the ample liquidity of those policy measures reduces the willingness to make reforms and encourages excessive risk-taking. The latter has the impetus to create growing public debts and subsequently inflation.

Either way, there is a distinction between monetary and economic policy in the Treaty.[2] The remit of the ECB is to conduct monetary policy, according to Article 282(1) and (4) TFEU. The primary goal is to maintain price-stability according to Article 127(1) TFEU. Despite the fiscal nature of unconventional policy, the ECJ gives the ECB broad discretion in pursuing the mandate without conducting a proportionate review (BVerfG 2020).

In respect of economic policy, the ECB shall support the general outcome so long as price-stability is not threatened. Yet the monetary mandate is limited by Articles 119 and 125 TFEU, which define the responsibilities of the Member States. Although the treaty demands coordination of economic policy in Article 119(1) TFEU, it is limited to the overall objectives of the EU according to Article 3 TEU. Furthermore, the treaty explicitly prohibits monetary interference in fiscal and economic matters, according to Article 119(2) TFEU and Article 2 of the ECB Statute.

Notwithstanding, monetary instruments are pivotal to the treaty provisions too. For instance, the direct purchase of government bonds is prohibited according to Article 123(1) TFEU (ECJ 2012, 2015, Rn 42). However, the ECJ reduces those restrictions significantly by arguing that a monetary measure cannot be treated as equivalent to an economic instrument for the reason that it may have indirect effects on the real economy.[3] Economically, this legal interpretation casts doubt on the demarcation line between economic and monetary policy. Indeed, this reading blurs the monetary mandate. In consequence, both verdicts would justify an almost illimitable utilisation of monetary policy in the European Monetary Union's (EMU) future.

Economists argue that unconventional policies over a prolonged time endorse fiscal profligacy. As a by-product, the central bank is ensuring the composition of the Eurozone without having the democratic legitimacy to do so. Hitherto, the Lisbon Treaty contends that the composition of the Eurozone is incumbent upon both political and democratic legitimation. This article argues that the verdicts focus on a too narrow legal vantage point and the democratic and philosophical boundaries are insufficiently considered.

The article is organized as follows. In Section 2, we expound both monetary programmes and the respective verdicts. Section 3 explains the inner workings of the ECB for legal scientists. Section 4 elaborates on the political involvement of the ECB since the onset of the global financial crisis. In Section 5, we substantiate the flaws about the OMT and PSPP. Section 6 uncovers two corroborating blemishes beyond economics. Overall, this article discloses a unique understanding about the ECB and the respective ECJ rulings.

## 2. Literature: Genesis of the OMT and PSPP Programmes

During the sovereign debt crisis, the Eurozone faced continuously rising bond yields in selective Member States. In a nowadays-famous speech on 26 July 2012, the ECB president Draghi (2012) announced:

> Within our mandate, the ECB is ready to do whatever it takes to preserve the euro. And believe me, it will be enough.

Subsequently, bond spreads declined significantly without the provision of any liquidity (Figure 1).

---

[2]   This holds despite the principle of conferral in Article 5(2) TEU.

[3]   The ECJ argues similarly, otherwise the ECB would no longer be in a position to fulfil its mandate. Economists note, however, that unconventional instruments intensify the misallocation of capital and impair the treaty provision of an open market economy in Article 119 TFEU.

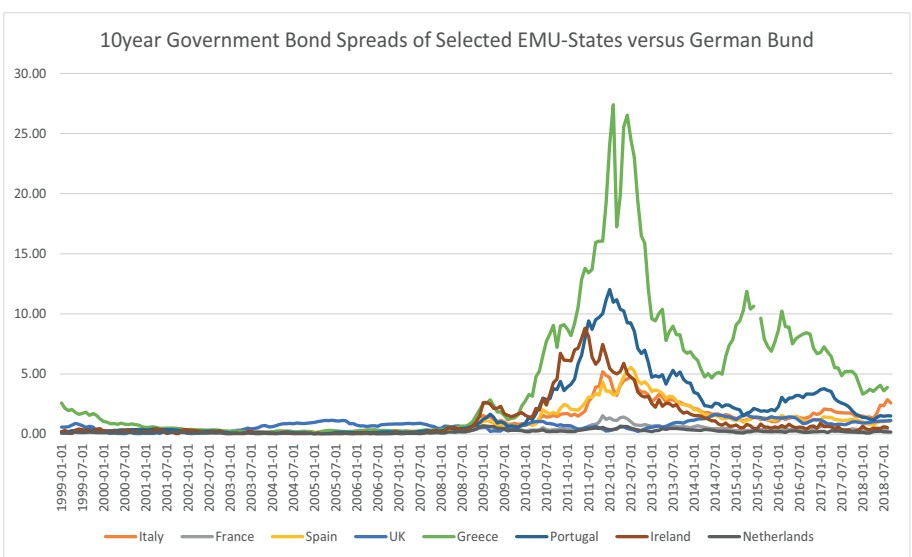

**Figure 1.** Government Bond Spreads in Eurozone.

On 6 September 2012, the ECB's Governing Council approved the formal criteria of Outright Monetary Transactions (OMTs) programme in order to reaffirm Draghi's commitment in July (Degenhart 2015; Siekmann and Wieland 2015). The formal requirements are threefold. First, bond purchases are in the secondary market and they are of limited size in respect to short-term maturities of one to three years with a time delay between the emission and purchase date. Second, the eligible Member States are under the European Stability Mechanism (ESM) and follow the herewith-necessary conditionality. Third, the ECB accepts the equal treatment to private creditors (i.e., pari passu) and commits to the sterilization of the new liquidity (Tuori 2016). On 14 January 2014, the German Constitutional Court requested an assessment of OMTs with Union law because the court sees a connection to domestic economic policy (BVerfG 2014, Rn 1–24). However, national courts are bound to a European-friendly interpretation of the ECJ.

On 16 June 2015, the ECJ published the verdict. It states that OMTs are consistent with European law, particularly to suppress the defected monetary transmission mechanism in order to guarantee the singleness of the monetary union (ECJ 2015, Rn 66, 67). Moreover, the ECJ argues that OMTs do not infringe upon the prohibition of monetary financing in Article 123(1) TFEU so long as the above requirements are satisfied (ECJ 2015, Rn 106, 115–9).

By 2012, in the Pringle v Ireland verdict, the ECJ argued that the ESM has no connection to economic policy either (ECJ 2012). Hence, the ESM does not infringe upon the no-bailout principle in Article 125 TFEU. Given that OMTs economically complement the ESM, two things stand out: on the one hand, OMTs provide further liquidity. On the other, they amplify the economic policy interference. In that regard it is rather remarkable that the ECJ concludes that the combination of OMTs and the ESM does not infringe Article 123 TFEU, although it disturbs the ESM conditionality by further liquidity. Therefore, on 21 June 2016, the German Constitutional Court enshrined even stronger domestic requirements on OMTs. Furthermore, the German court argues that the ECJ's interpretation is a carte blanche to all monetary instruments (BVerfG 2020; Herzog 2020b).

Likewise, on 22 January 2015, the ECB decided to initiate a new large-scale asset purchase programme, the so-called Public Sector Purchase Program (PSPP).[4] Under the PSPP, the ECB purchases outright eligible debt securities. The PSPP was justified by the ECB due to increasing downside risks to price developments, lower than expected monetary stimuli, downward drift in economic indicators, and the fragmentation of monetary transmission. The initial monthly purchase of the PSPP amounted to 60 bn euro and increased to 80 bn euro in 2016. In total, the asset volume has accumulated to almost 2.6 tn euro as of January

---

4    Decision (EU) 2015/774 of the European Central Bank of 4 March 2015 on secondary markets public sector asset purchase programme.

2019.[5] The PSPP programme stopped on 31 December 2018, although the ECB continues to reinvest the principal payments of the securities purchased, as long as it deems necessary. Since September 2019, the ECB has restarted the program.

It is important to note that the PSPP programme is different to OMTs and has special features. First, it purchases only marketable and eligible[6] assets in the secondary market with restrictions on the total monthly volume. Second, countries in an ongoing assistance programme, such as Greece, are suspended from the programme. Third, the debt securities must have a minimum remaining maturity of one year to a maximum of 30 years. Fourth, there is no purchase of newly issued securities ("blackout period"). Fifth, under the PSPP the ECB purchases 10 percent of assets by international organisations and 90 percent by governments of its own jurisdiction. Lastly, 90 percent of asset purchases shall be made by national central banks and only 10 percent by the ECB.

Once more, on 18 July 2017, the German Constitutional Court requested a preliminary ruling under Article 267 TFEU from the ECJ. The Court asked (i) does the PSPP fall within the scope of the ECB's mandate, as defined in Article 127(1) and (2) TFEU? (ii) Does the purchase infringe upon Article 123 TFEU? (iii) Does the PSPP infringe upon the principle of democracy?

On 11 December 2018, the ECJ ruled first that the PSPP programme does not exceed the ECB's mandate and second that the PSPP programme does not infringe the prohibition of monetary financing (ECJ 2018). The European Court emphasises that the purchase of government bonds on secondary markets is not equivalent to a purchase of bonds on the primary market and does not reduce the impetus of Member States to follow sound budgetary policy. Unsurprisingly, for the first-time in German legal history, the German Constitutional Courted (Bundesverfassungsgericht) has rejected the ECJ's assessment of the PSPP by utilising the ultra-vires concept (BVerfG 2020; Herzog 2021).

### 3. Working of the European Central Bank

In order to understand the disagreement between the economic and legal assessment, we briefly describe the working of monetary policy. The ECB is a unique central bank, and unlike the US Federal Reserve, it does not have a single government as a counterpart. Like other modern central banks, it uses open market operations as a standard tool. The ECB is managed by the Governing Council. The decision-making is by a process of consensus largely built upon the president. Given that the executive board is appointed for a single eight-year term, the ECB is arguably the most powerful and independent central bank in the world (Arnone and Romelli 2013; Brunnermeier 2012; Eijffinger 2006).

It is noteworthy that the European treaty reveals an unusual monetary philosophy. Article 127(1) TFEU states that the primary mandate is to maintain price stability in conjunction with Article 120 TFEU.[7] Within that framework, the ECB defines and executes monetary policy independently. However, the ECJ points out that the Treaty contains no precise definition of monetary policy but only defines the objectives and instruments available (ECJ 2018). Furthermore, an unspoken monetary philosophy became apparent during the European sovereign debt crisis. The ECB executes policy in a way through which it preserves the singleness of the Eurozone. Of course, any exceptional decision of the ECB was always justified by Article 127 TFEU.

There is broad consensus in economics that direct purchases of bonds are not the remit of a central bank due to the threat of moral hazard and fiscal dominance, which harms central banks' independence (Draghi 2012). Thus, one must construe the ECB's mandate narrowly and in conjunction with Article 123 TFEU and Article 125 TFEU.

---

5    ECB, available: www.ecb.europa.eu/mopo/implement/omt/html/index.en.html (accessed on 15 December 2018).

6    ESM—Article 5(1) and 5(2) EU2015/774.

7    More generally, Article 3(1)(c) TFEU states that the Union is to have exclusive competence whose currency is the euro. In conjunction with Article 282(1) and (4) TFEU, the ECB conducts the monetary policy of the Union and adopts the measures necessary to carry out its task defined in Articles 127 to 133 TFEU.

To better understand the economic rationale, we outline the inner workings of monetary policy. First, one has to appreciate the monetary transmission mechanism. The transmission mechanism links the aim of price stability in Article 127(1) TFEU with the interest rate. Unsurprisingly, there is a nontrivial relationship between prices and interest rates due to endogenous and exogenous forces. Endogenous factors include expectations in wage- and price-setting. Exogenous features are commodity prices, fiscal policy or the global economy. As a matter of fact, the fiscal exposure of the 19 Member States makes monetary policy more challenging in the Eurozone than in the US. Particularly the last issue explains why monetary policy has to avoid a broader interference to economic policy, particularly in a monetary union (Beetsma and Bovenberg 2003). Otherwise, the central bank creates uncertainty, erratic impulses and a misallocation of capital (Bernanke 1983; Brainard 1967; Rudebusch 2001). Indeed, establishing equal financing conditions in a monetary union is a transgression of the monetary mandate.

The main interference comes from the monetary instruments. The most basic instrument is an open-market operation. The implementation is done through a repurchase agreement, in short 'repo'. A repo is a transaction by the central bank made against collateral assets. When a repo expires, the commercial bank pays back its loan and regains the collateral. Through these liquidity operations, the ECB influences the prevailing short-term interest rate, while respecting Article 127 TFEU aligned with Article 120 TFEU.

Two points stand out. First, normally the ECB does not own or purchase the collateral. Second, the ECB strictly limits its operational risk by holding collaterals. Yet, under unconventional operations, such as OMTs or PSPP, the ECB does purchase assets. In addition, under normal conditions the ECB reduces the risk of default by demanding a so-called haircut in respect to the collaterals, meaning that the loan value is lower than the value of the collateral. However, this principle does not apply to the OMTs and PSPP programmes either.

Furthermore, in normal times, eligible collaterals are triple-A-rated assets. During the crisis, the eligibility was reduced to single-A and triple-B. In May 2010, the ECB even removed any minimum rating for Greek government bonds. Later in 2011, they followed with this exception for Irish and Portuguese bonds. In special cases the ECB now provides funding even without eligible collaterals by way of Emergency Liquidity Assistance (ELA).[8] In 2015, ELA funding became a powerful tool to finance distressed banks in vulnerable countries.

Article 18.1 of the ECB Statute does not preclude the possibility of ELA or bond purchasing programmes as long as the programme supports the general economy, as laid down in Article 3 TEU and as long as it is contingent under Article 127(1) and Article 282(2) TFEU (ECJ 2018). The latter contingency is, however, critical. So far, the ECB has no rule for selling bonds before maturity under OMTs or PSPP.[9] Consequently, buying assets outright is a part of the monetary mandate but only to the extent that the instrument is proportionate to the general objective and does not interfere with other policies (BVerfG 2020).

## 4. What Kind of Central Bank Is the ECB?

This section explores how the unconventional instruments created an unprecedented degree of political involvement. Since the global financial crisis, monetary policy has changed in two dimensions. First, there has been an extension of conventional operations. Second, central banks utilised new unconventional instruments. The evolution of conventional versus unconventional operations tells us what kind of central bank the ECB is in practice. There are two crucial delimitations of conventional versus unconventional instruments.

Firstly, under unconventional monetary policy, the central bank directly purchases assets, while under conventional operations an asset takes the form of collateral. Of course, purchasing government bonds in the secondary market provides direct liquidity only to

---

[8]  Technically, this is like a repo with three exceptions (Article 14.4 ECB Statute). First, under ELA low-quality collateral is sufficient. Second, the risk of ELA stays at the national central banks. Third, the ELA funding is usually more expensive for banks. The ECB's Governing Council must approve each ELA with a two-thirds majority.

[9]  This follows from Article 119(2) TFEU and Article 127(1) TFEU, read in conjunction with Article 5(4) TEU.

the banking system; however, this liquidity is forwarded directly to governments in the first place.

Secondly, conventional operations carry little risk due to haircuts under the collateral policy. In contrast, unconventional policy carries default risks inasmuch as the holding of bonds is until maturity. The latter exposes the central bank to new risks of fiscal and financial dominance (Brunnermeier 2018). This means that the central bank gets strongly contingent on the conduct and quality of public policy and commercial banking. Hence, the German constitutional court argues in a pertinent and persuasive manner that unconventional operations are only acceptable if they are not be beyond the mandate, suitable and follow the proportionality principle in Article 5 TEU (BVerfG 2020).

In 1873, Walter Bagehot explored the need for lender-of-last-resort policies during financial turmoil (Bagehot 1873). Today, the provision of liquidity is a core feature of modern central banking. We distinguish three phases throughout the global financial crisis. In the first phase from 2007 to 2008, the ECB provided moderate liquidity to solvent banks by using fine-tuning operations. In summary, the first phase fully retained the monetary mandate. In the second period of 2008 to 2011 the ECB devoted additional liquidity via long-term refinancing operations (LTROs) and targeted-LTROs (Herzog 2016). In consequence, the second phase did not go beyond the mandate. Yet the third phase is a tipping point. At the end of 2011, the ECB utilised, for the first time in monetary history, unorthodox unconventional operations. This period created a new world of instruments, such as ELA liquidity or large-scale asset purchase programmes. The unconventional operations triggered vigorous publicity and controversy not limited to Europe (Konrad 2013; Tuori 2016; Taylor 2016, 2017).

The starting point was the contagious sovereign debt crisis in Greece. Indeed, the ECB was the only game in town, given the slow decision-making processes due to unanimity in the EMU's architecture. Nonetheless, the ECB has no legal power to ensure the singleness of the Eurozone—Member States are primarily responsible. For instance, the ECB rendered Greek bonds as adequate collateral despite a joint rescue programme by Member States and the IMF in April 2010 (Herzog 2012; ECB 2011; IMF 2010, 2017).[10] Furthermore, the ECB extended the provision of liquidity via ELA despite default risk. According to Article 14.4 of the ECB Statute, the provision of emergency liquidity assistance is under the control of the ECB, but the execution and risk is at the national central bank. This tool assists prima facie solvent banks with a lack of eligible collateral. During the heyday of the crisis, Greek banks would have been unable to obtain liquidity under normal central bank rules.

In January 2015, Alexis Tsipras, the Greek prime minister, refused the conditionality of the ESM programme and called a referendum (Martin 2012). On 4 February, the ECB's Governing Council decided to lift the waiver, which previously allowed Greek bonds as collateral in monetary operations. Thus, the ECB exposed Greek banks to the more expansive ELA funding. Note, ELA loans have a higher interest rate, although not published, which is somewhere in the range of 100 to 175 basis points (Bindseil and König 2012; Jones 2015). In addition, each ELA loan requires approval by the Governing Council. As a consequence of this policy, the central bank increased both economic and political pressure to Greek banks and the government. The Greek government faced the dilemma of either accepting the new rescue programme with even stronger conditionality or risking a collapse (Boland and Spiegel 2014). This type of ultimatum is controversial because it is enforcing political pressure to democratic governments.[11] This infringes the principle of democracy—particularly in a monetary union.

In essence, the ECB became a quasi-fiscal player (Brunnermeier 2018, p. 336). Similarly, Mankiw and Reis argue that the central bank is now a fiscal agent and the ECB should not have the right to put political pressure on Member States (Hellwig 2015; Mankiw and Reis 2018). Many economists have perceived the third phase as a slippery slope and a dangerous precedent (Sinn 2012; Herzog and Ferencz 2019; Weidmann 2011, 2012).

---

[10] At that time, the IMF, the ECB, and others assessed the Greek debt as vulnerable (Wolf 2012).

[11] The ECB put pressure on Ireland to use taxpayers' money to bail out Irish banks (Report on Irish Crisis, 2016).

The same procedure occurred in the case of Cyprus.[12] First, the ECB forced banks to accept ELA funding. Second, the ECB put political pressure on the government in order to agree upon a political rescue programme. The ECB press release reveals this strategy:

> The Governing Council of the European Central Bank decided to maintain the current level of Emergency Lending Assistance (ELA) until Monday, 25 March 2013. Thereafter, Emergency Lending Assistance (ELA) could only be considered if an EU/IMF program is in place that would ensure the solvency of the concerned banks (ECB 2013).

It is hardly the mandate of any central bank, even in difficult times, to pull governments into a rescue programme. It is remarkable that the ECB no longer accepted Greek or Cypriot bonds as collateral after European political opposition or an election of a non-mainstream government. Is this behaviour a sign of political involvement or independence?

First, de facto, the ECB is independent. Yet, it is a member of the troika—now called 'institutions' —that assesses domestic policy under the ESM programme. Hence, the central bank is now becoming a political player too. Second, the ECB enforces unconventional policies, as explained above, in order to pull countries into rescue programmes. This policy reveals a sign of political involvement. In fact, the ECB has been amplifying its political involvement even further. For instance, the four-year economic strategy of the Irish government was coordinated with the ECB. This reveals a hidden blemish and establishes a murky interference with domestic policy (Lombardi and Moschella 2015).[13] Indeed, this demonstrates a new philosophy of the ECB.

Apart from the political involvement, there is a further critique. In 2010, under the Securities Markets Program, the ECB commenced the purchase of bonds with a total volume of 220 bn euro until early 2012. The central bank justified this programme with the fragmentation of markets. On the one hand, acquiring sovereign bonds is riskier due to the risk of default (Uhlig 2015). On the other hand, the repatriation (sterilisation) of liquidity works if and only if the ECB finds investors willing to buy the assets in future. This exposes the ECB to value and time risk as well as fiscal and financial dominance. The latter risk endangers the monetary independence too. In summary, unconventional operations expose the monetary mandate to a higher degree of political involvement and subsequently lower independence.

This critique is particularly amplified under the OMT programme. At first, OMTs support only indebted stand-alone states and not the Eurozone as a whole. Second, OMTs exert political pressure because the prerequisite is membership in the ESM programme. Hence, OMTs corrode monetary independence and create moral hazard. Similarly, the critique applies to the PSPP; here it is the open-ended nature on the one hand and the risk sharing on the other.[14]

In addition, there is a philosophical difference between the OMT and PSPP programme. While under OMT, the ECB buys bonds of single indebted Member States, the ECB excludes single indebted Member States from the PSPP. Hence, the ECB violates the principle of equal treatment, which is key for a supranational independent institution. Indeed, both programmes reveal a vicious circle for an independent central bank with a primary mandate of price-stability (BVerfG 2016, Rn 130). For instance, the US Federal Reserve has a strict prohibition on purchasing any public bonds of the 50 US States (BVerfG 2016, Rn 89, 155). Despite empirical evidence on the risks, it is a mystery why the ECJ does not consider these issues. In summary, the third phase put the ECB in a new position:

> The ECB appears both as a hero and a victim of the crisis (…). As an institution, however, it suffered collateral damage. (…) despite the power shift away from EU institutions, the ECB is the only one that grew in power. (…)

---

[12] There are several examples: ELA to bailout Hypo Real Estate in 2008, Irish banks in 2010, Cypriot banks in 2013, and Greek banks in 2012 and 2015.

[13] Many economists show that in other monetary unions such as the United States, bailouts have been rigorously excluded for more than 150 years. President Obama, for instance, refused to bail out California in 2009. The US has a credible no-bail provision.

[14] Indeed, the PSPP increases the ECB's balance sheet by 1266 percent.

but in becoming more powerful, the ECB also had become more vulnerable. An independent central bank just seemed to be making too many decisions (Brunnermeier 2018, p. 374).

Issing (2017), the well-respected former ECB chief economist, reiterates that unconventional operations have not, and will not, solve the root-causes of the Greek crises. In addition, at the World Economic Forum in Davos, Rogoff (2020), an international expert in monetary economics said the ECB's policy does not fix the problem and the out-of-the-box policy blows up the economy.[15] All testimonials corroborate the new nature of political involvement under unconventional policy in the Eurozone.

## 5. Further Blind Spots of Legal Analysis

In the following section, we overview the economic notions that are also partly unrecognised under the legal analysis.

### 5.1. Moral Hazard

Any monetary intrusion to domestic policy is questionable, if not naïve. Suppose the government follows the requests of the central bank. Under this assumption, the government loses power over public policy, which corrodes its democratic legitimacy. Otherwise, not following the monetary dictum threatens the country's stability. Both outcomes demonstrate a hidden agenda of unconventional policy. It is a kind of monetary imperialism and rather immoral for an independent institution.

If Member States tolerate this interference, the central bank creates a self-fulfilling prophecy. This prophecy is an implicit bailout, as illustrated by the quote: «whatever it takes». Since the US stock market crash in 1987, the literature calls this prophecy a central bank put option.[16] This wording refers to the US Federal Reserve chair Alan Greenspan, who bailed out weak banks by liquidity and lower rates. However, this commitment erodes independence, particularly in a monetary union. Independent central banks do not have a mandate to turn a bad equilibrium into a good one—only to stabilize an equilibrium. Yet, in doing so, central banks create moral hazard. As a result, countries gamble for resurrection, which endangers the long-run stability of the Eurozone (Hellwig 1995).

### 5.2. Redistribution

Research exhibits a side effect of unconventional operations on inequality. Similarly, studies find that it is hard to escape the conclusion that unconventional policy provides an indirect subsidy to asset holders (Siekmann and Wieland 2015). Indeed, large-scale direct purchases of assets lead to a redistribution effect (Wyplosz 2015). This shifts the borderline between economic and monetary policy and blurs the monetary mandate. Indeed, one can observe the redistribution by the so-called TARGET2 imbalances across the Eurozone (Fahrholz and Freytag 2012). In this regard, TARGET2

come[s] close to constituting hidden transfers, directly to benefiting some Member States (. . . ) and some financial institutions (. . . ), while indirectly benefiting creditors of (. . . ) Euro area core countries (Tucker 2015; Tuori and Tuori 2014).

However, an independent central bank with a mandate of price-stability ought to avoid this policy effect.

### 5.3. Infringement of No-Bailout Article 125 TFEU

Public policies are within the sovereignty of Member States. Hence unconventional policy, which deliberately redistributes, does not belong to the ambit of an independent central bank (Tuori 2016). The policy interference and the broad discretion of the ECB's

---

[15]  https://www.weforum.org/events/world-economic-forum-annual-meeting-2020/sessions/escaping-the-liquidity-trap (1 February 2020) and https://www.weforum.org/events/world-economic-forum-annual-meeting-2020/sessions/no-safe-asset (1 February 2020).

[16]  Economists call it a 'Greenspan-put' (Carlson 2007).

mandate is a problem rather than a solution (ECJ 2018, Rn 58, 59). In both rulings the ECJ does not follow the principle of proportionality, which means it must be suitable for attaining the objective and should not go beyond what is necessary (BVerfG 2020).

If a monetary program is designed to prevent a single-country break up in the Eurozone, the proportionality principle comes into question. Does one country justify an unorthodox OMT operation to the disadvantage of other sovereign Member States? A single-country break up is—what else—contingent on a sovereign decision of each Member State. The central bank has no remit to keep a single country inside or outside the Eurozone. The ECJ broad interpretation of the monetary mandate is a veiled bailout and breaches the principle of proportionality (BVerfG 2020).

### 5.4. Infringement of Prohibition of Monetary Funding Article 123 TFEU

Article 123(1) TFEU prohibits the central bank from granting overdraft facilities or any other type of credit facility to public authorities (ECJ 2015). Nonetheless, the ECJ approves unconventional programmes as long as the purchase is in the secondary market (ECJ 2018). Yet, the ECJ applies a very narrow and formalistic reading of Article 123 TFEU (ECJ 2018, Rn 111, 131f). Strategically, one can argue that the ECB is trapped with 19 Member States in a game of chicken. Figuratively speaking, two drivers are heading for a bottleneck and both want the other to give in first. Each of the two parties hopes that the other will be the first to react and bear the negative consequences. This problem occurs because the treaty does not contain a precise definition of monetary policy (ECJ 2015, 2018, Rn 42, Rn 50). Yet imprecise Treaty provisions do not justify any monetary operation without a legal proportionality review. The ECJ concludes:

> In the light of those considerations, it is apparent that a programme such as that announced in the press release, in view of its objectives and the instruments provided for achieving them, falls within the area of monetary policy. The programme is contributing to the stability of the euro area, (...) [even if it] interferes in economic policy (ECJ 2015, Rn 45).

In summary, unconventional operations are partly beyond the confines of the monetary realm. However, the broad and fuzzy mandate is a carte blanche to the Eurosystem.

## 6. Beyond the Economists' Toolkit

Finally, we unravel two further notions that are beyond economics, yet relevant to a comprehensive legal analysis: on the one hand, the role of democratic legitimacy (Scharpf 1999); on the other hand, the philosophical misconception of the power of liquidity (Sandel 2012).

### 6.1. Democratic Legitimacy

The former ECB chief economist Otmar Issing conducted a compelling thought experiment about the democratic legitimacy of unconventional policy. He argued that unconventional policy is effective if and only if the government sticks to the commitment. However, a domestic reform always requires a democratic vote. Hence, it is naïve to believe that Member States will adhere to the omnipresent central bank's request—Greece is the best example (Beetsma and Bovenberg 1999, 2003; Wyplosz 2015; Issing 2017). Empirical evidence corroborates the flawed conclusion of the ECB (Cukierman 2008; Martinez Garcia 2017; Masciandaro and Romelli 2015).

A more sophisticated issue is that there is a deeper problem with output-oriented legitimacy, which denotes a democratic response to any policy output because an independent institution is neither fully responsible nor punishable for exogenous and endogenous outputs (Scharpf 2011). Similarly, unconventional policy affects input-oriented legitimation. That means any policy action triggers trade-offs, such as painful sacrifices made in the pursuit of benefits (Schmidt 2006, 2012). Under the law of democracy, these policy choices must be free. Yet under OMTs, countries lose the free choice as the troika must approve all policies (Kydland and Prescott 1977; Scharpf 2011). In summary, it is naïve to assume that independent central banks do have the mandate to request binding reforms

from democratic Member States. Indeed, central banks have less to gain and more to lose under unconventional policies.

*6.2. Philosophical Misconception*

Additionally, there is a profound misconception about the commitment of unconventional operations. Very early economists discovered that mainstream economics overlooks the 'commercialization effect' (Hirsch 1978). One can find empirical evidence that money crowds out the intrinsic commitment of reforms, or even corrodes it (Sandel 2013). In some cases it might even lead to policy choices for the wrong reason. In addition, it is particularly dangerous that Western liberal democracies are becoming more and more aligned to technocratic or populist governance (Herzog 2020a). This creates a political sphere vacant of free speech and rational choice (Herzog 2020c).

Several studies have confirmed the power of markets and the limits of money (Ariely et al. 2009; Bénabou and Tirole 2003, 2006). There is evidence that extrinsic incentives impair intrinsic motivation. In certain situations, money even creates the feeling of bribery. Research work shows that people donate more blood if they do so voluntarily or in general they act decisively if there is an intrinsic motivation (Falk and Szech 2013; Gneezy and Rustichini 2000; Mellström and Johannesson 2008; Titmuss 1971).[17]

Equally there is evidence that countries with high default risk lack a credible intrinsic commitment and gamble for resurrection (Reinhart and Rogoff 2009, 2011; Hellwig 1995). Requesting reforms in return for funding under the OMT programme mirrors the findings in the economic literature. Indeed, countries with intrinsic motivation impose more sustainable and effective reforms.[18] Hence, if the central bank utilises unconventional policy, it is conveying moral considerations and accepts an undemocratic commercialization of public policy (Arrow 1972; Fryer 2011). From this vantage point, a central bank becomes a technocratic institution that puts a price tag on public policy (Gneezy and Rustichini 2000; Volpp et al. 2009).

## 7. Concluding Thoughts

The economist Alan Meltzer once said 'capitalism without failure is like religion without sin.'[19] This article establishes that the European Court and European union law are not exempt from epistemology. Even the ECJ confesses that

> (...) monetary policies are usually of a controversial nature (...). [Further] (...) any policy instrument should be assessed in respect to the threat of domestic interference and the related liability risks (...) (ECJ 2015, Rn 75,125f.).

It remains to be seen when the European Court of Justice will reorganise its blemishes and adapt to a consistent and robust notion. In the meantime, it is best for central bankers to remain humble in what they aspire to achieve.

---

[17] Further: Heyman and Ariely (2004); Holmas et al. (2010); Janssen and Mendys-Kamphorst (2004).

[18] They stick to painful commitments due to intrinsic virtue. Examples are Ireland and the UK in the 1980s. Both countries made painful social cuts due to high unemployment and a weak economy. In Germany in the year 2000, after imposing reforms called AGENDA 2010, Chancellor Gerhard Schroeder changed the long-run growth trajectory. Despite violent protests across the country, German politics adhered to the intrinsic reform commitment. In return the German economy benefited and is now the powerhouse of Europe. Reference: Economist. The sick man of the euro: the biggest economy in the euro area, Germany's, is in a bad way. Special Report, 3 June 1999.

[19] New York Times, 12 May 2017. www.nytimes.com/2017/05/12/business/economy/allan-h-meltzer-dead-conservative-economist.html; retrieved 15 May 2017.

**Funding:** The article processing charge was funded by the Baden-Württemberg Ministry of Science, Research and Arts in the funding programme Open Access Publishing.

**Institutional Review Board Statement:** Not applicable.

**Informed Consent Statement:** Not applicable.

**Data Availability Statement:** Not applicable.

**Acknowledgments:** I thank Christian Calliess for valuable discussions on related topics and the RRI–Reutlingen Research Institute for administrative and technical support of my research.

**Conflicts of Interest:** The author declares no conflict of interest.

## Abbreviations

The following abbreviations are used in this manuscript:

| | |
|---|---|
| ECJ | European Court of Justice |
| ECB | European Central Bank |
| BVerfG | Bundesverfassungsgericht/German Federal Constitutional Court |
| TEU | Treaty of the European Union |
| TFEU | Treaty of the Function of the European Union |
| Rn | Randnumber/Reference number |
| OMT | Outright Monetary Transaction |
| PSPP | Public Sector Purchase Programme |

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
