# Peer review of "Hidden Blemish in European Law: Judgements on Unconventional Monetary Programmes"

_laws, 2021_

Round 1

Reviewer 1 Report

A very interesting discussion that is well constructed. The literature and data support the contentions. The argument is well developed and is rational. The article make a worthhile contribution ot the knowledge on the topic.

Author Response

Dear Referee No 1,

thank you very much for your positive paper feedback and good comments.

Reviewer 2 Report

The paper submitted is appealing and presents some critical considerations on the role of the European Central Bank and monetary policies, following two judgments of the ECJ.

I have no suggestions other than to correct a few reported typos: line 39, 107, 123, and 452. The paper seems to have a good structure.

Author Response

Dear Referee No. 2,

thanks a  lot for you positive and good comments.

Of course, I have corrected all indicated typos and textual errors in the respective lines in my final manuscript. I really appreciated your feedback in order to improve my article.

Reviewer 3 Report

NIce article. I do not agree with your critical stance against the ECJ. I feel, the requests you voice in n. 466 ss are simply not feasible. Nonetheless, your manuscript is a valuable contribution to an important discussion.

Author Response

Dear Referee No. 3,

thanks a  lot for your valuable feedback and comment.